# Combined Silymarin and Cotrimoxazole Therapy Attenuates Pulmonary Fibrosis in Experimental Paracoccidioidomycosis

**DOI:** 10.3390/jof8101010

**Published:** 2022-09-27

**Authors:** Victor Quinholes Resende, Karoline Hagata Reis-Goes, Angela Carolina Finato, Débora de Fátima Almeida-Donanzam, Amanda Ribeiro dos Santos, Jonatas Perico, Barbara Casella Amorim, James Venturini

**Affiliations:** 1Faculdade de Ciências, Universidade Estadual Paulista (UNESP), Bauru 17033-360, SP, Brazil; 2Faculdade de Medicina, Universidade Estadual Paulista (UNESP), Botucatu 18618-970, SP, Brazil; 3Faculdade de Medicina, Universidade Federal de Mato Grosso do Sul (UFMS), Campo Grande 79070-900, MS, Brazil

**Keywords:** paracoccidioidomycosis, pulmonary fibrosis, pulmonary sequelae, silymarin, cotrimoxazole

## Abstract

Paracoccidioidomycosis (PCM), which mainly affects rural workers, is a systemic mycosis caused by the *Paracoccidioides* genus that induces pulmonary sequelae in most adult patients, causing serious disability and impairing their quality of life. Silymarin is herbal medicine with an effective antifibrotic activity. Considering that in PCM, antifibrotic treatment is still not available in pulmonary fibrosis, we aimed to evaluate combined silymarin and cotrimoxazole (CMX) therapy via the intratracheal route in BALB/c mice infected with *P.* *brasiliensis* yeast. After 12 weeks of treatment, the lungs were collected for the determination of fungal burden, production of OH-proline, deposition of collagen fibers, pulmonary concentrations of cytokines, and expression of fibronectin, α-SMA, MMP-2, MMP-9, and TIMP-2. Spleen cell cultures were also performed. Our results showed that infected mice treated with combined silymarin/CMX showed lower deposition of collagen fibers in the lungs and lower pulmonary concentrations of hydroxyproline than the placebo groups. Decreased levels of TGF-β1 and fibronectin and high levels of MMP-2 and IFN-γ were also observed in this group of mice. Collectively, our findings indicate that the combination of antifungal treatment with silymarin has a potent antifibrotic effect associated with an immunomodulatory effect that potentializes the antifungal immune response.

## 1. Introduction

Paracoccidioidomycosis (PCM) is a systemic mycosis caused by thermodimorphic fungi of the genus Paracoccidioides [1,2]. Restricted to Latin America, the disease has a heterogeneous distribution, with areas of high and low endemicity. The highest incidence of the disease occurs in Brazil, Argentina, Colombia, and Venezuela, with 80% of the registered cases in Brazil [3].

The main clinical forms of the disease are acute/subacute (AF) and chronic (CF) forms. AF, or the juvenile form, mainly affects children, adolescents, and young adults; it involves the lymph nodes, spleen, liver, and bone marrow. On the other hand, CF, or the adult form, affects adults over 30 years of age, and it mainly involves the lungs and mucosa of the upper aerodigestive tract [4]. Almost all patients with CF present with pulmonary sequelae after treatment of PCM, with subsequent devastating impact on the individual’s work productivity and quality of life [5].

In recent years, our group has focused on the study of pulmonary sequelae in patients with CF-PCM, and our findings have shown pro-fibrotic alterations since diagnosis along with prolonged systemic and low-grade pro-inflammatory profiles during and after a complete and successful antifungal therapy [6,7,8,9,10]. In addition, we previously established a murine model of pulmonary PCM, in which BALB/c mice presented with intense pulmonary fibrosis (PF), and we evaluated antifibrotic drugs and combined antifungal therapies [11].

A pharmacological treatment for pulmonary sequelae in patients with PCM is not yet available; however, experimental studies have identified promising therapeutic strategies for treating PF in PCM, including drugs used for idiopathic PF, immunotherapy, antibody-based therapy, cellular therapy, and vaccination (review in [12]). In the present study, we proposed a new therapeutic approach for PF in PCM that includes herbal formulations and bioactive plant molecules as potential candidates.

Silymarin, an extract of *Silybum marianum* L. (milk thistle), has a remarkable biological effect [13] and an antifibrotic role in liver fibrosis [14]. Silymarin is commercially available as Legalon^®^, Silimalon^®^, Siliphos^®^, and other tradenames. It is a safe and well-tolerated herbal medicine [15] that has been evaluated in 65 registered studies in the ClinicalTrials.gov database (https://www.clinicaltrials.gov/ct2/results?cond=&term=Silymarin&cntry=&state=&city=&dist=, accessed on 15 June 2022). The beneficial effects of silymarin are due to the presence of flavonoids, in which the most prevalent component is silibinin (50–70%), and it is the most active phytochemical that is responsible for the beneficial effects of the extract. In addition, other flavonolignans are also present, such as silichristin (20%), silidianin (10%), isolibinin (5%), and a few flavonoids [16,17].

Bannwart et al. [18] previously reported the promising role of silibinin as a useful alternative for overcoming the deleterious effects of systemic inflammation in PCM. The authors demonstrated that silibinin exerts anti-inflammatory and antifibrotic effects on CD14+ human monocytes challenged with a virulent strain of *P. brasiliensis* (*Pb*) by partial inhibition of p65NF-κB activation. Therefore, we aimed to investigate the effect of combined silymarin and cotrimoxazole (CMX) therapy as an antifibrotic treatment in a *Pb*-induced PF murine model.

## 2. Materials and Methods

### 2.1. Fungal Culture

*Paracoccidioides brasiliensis* (isolate 326, GenBank accession number: MH367529.1) isolated from a patient in Botucatu, SP, Brazil, was sub-cultured biweekly in GPY culture medium (2% glucose, 1% peptone, and 0.5% yeast extract) and incubated at 35 °C in the yeast phase. The viability of the fungus was monitored via staining with blue cotton lactophenol.

### 2.2. Mice Infection

Isogenic 8–12-week-old male BALB/c mice were obtained from the Lauro de Souza Lima Institute, Bauru, SP, Brazil. The mice were maintained under appropriate environmental conditions and received balanced feed and water ad libitum. Mice were intratracheally administered 10^6^ yeasts of *P. brasiliensis* (strain 326) in sterile buffered saline (SBS), whereas uninfected animals received only SBS. For the intratracheal administration, mice were anesthetized intraperitoneally with ketamine (80 mg/kg) and xylazine (10 mg/kg). This experimental model was previously standardized to induce intense PF [11].

### 2.3. Treatment Dosage and Administration

Silymarin (Silimalon^®^; Zydus Nikkho, Rio de Janeiro, Brazil) and combination of trimethoprim–sulfamethoxazole, also called cotrimoxazole (CMX; Bactrim^®^ oral solution, Roché, São Paulo, Brazil), were administered via oral gavage once a day in the morning with ad libitum food and water. The treatment dosages were as follows: CMX, 200 mg/kg body weight [4,11,19]; silymarin 100 mg/kg [20], in the form of an aqueous suspension of 1% carboxymethylcellulose (Sigma-Aldrich, St. Louis, MO, USA).

### 2.4. Experimental Design

Mice were randomly distributed into four groups (*n* = 7), according to combined therapies as follows: (1) combined silymarin and CMX treatment, (2) silymarin and placebo (SBS) combined treatment, (3) CMX and placebo (1% carboxymethylcellulose in sterile saline solution) combined treatment, and (4) placebo–placebo combined treatment. All treatments were initiated eight weeks after infection. At this time point, the lungs of *P. brasiliensis* (*Pb*)-infected mice showed large typical granulomas, were rich in viable fungi, and were surrounded by intense deposition of collagen fibers, as previously described [11]. After 12 weeks of treatment (20 weeks post infection), the mice were anesthetized by inhalation of isoflurane, followed by inhalation of carbon dioxide. In each set of experiments, successful inoculation was confirmed via histopathological evaluation. Failed inoculation was considered if no infectious injury was identified in the lung tissue; thus, the mice were not included in the analyses. All experiments were performed in duplicate, as recommended by the ethical committee. All procedures were performed in accordance with the ethical standards established by the Brazilian College of Animal Experimentation.

### 2.5. Histopathological Analysis

Lung fragments were removed and fixed in 10% buffered formalin. Paraffin-embedded sections (4 μm) were stained with hematoxylin and eosin (H&E) and Sirius red. Polarized light microscopy images were obtained using Sirius Red staining, and morphometric analysis was performed using the ImageJ software, version 1.48 (NIH, Bethesda, MD, USA).

### 2.6. Hydroxyproline Assay

Pulmonary hydroxyproline was photometrically measured in lung hydrolysates, as previously described [21]. Briefly, the samples were weighed, hydrolyzed with 6 N HCl at 110 °C for 16 h, filtered, mixed with methanol, and evaporated using a vacuum concentrator. Crystallized samples were dissolved in 50% isopropanol and incubated with 0.6% chloramine-T (Sigma-Aldrich) for 10 min. Ehrlich’s reagent (100 μL; Sigma-Aldrich) was added, and the samples were incubated at 50 °C for 45 min with constant shaking. Concentrations of total lung OH-proline were calculated against a standard curve. The results were normalized per gram of tissue.

### 2.7. Western Blotting

Protein quantification of the lung fragments was performed using the Pierce BCA Protein Assay kit (Thermo Scientific, Rockford, IN, USA). Thirty micrograms of protein lysates were fractionated using the Mini-Protean II electrophoresis system (Bio-Rad, Hercules, CA, USA) in a 12% polyacrylamide gel and transferred to a nitrocellulose membrane using the Trans-Blot Turbo Transfer System (Bio-Rad). Membranes were blocked with 5% BSA basal solution (10 mM Tris base, 150 mM NaCl, and 0.05% Tween 20) for 2 h at 25 °C and then incubated overnight with primary antibodies (1:1000 dilution) against fibronectin, including MMP-2, MMP-9, α-SMA, TIMP-2, and β-actin (Cell Signaling Technology, Danvers, MA, USA). The membranes were subsequently incubated with peroxidase-conjugated secondary antibody-HRP at a 1:80,000 dilution (Sigma-Aldrich) for 2 h at 25 °C. Then, the membranes were washed and incubated in a dark room with a luminol chemiluminescent substrate (Clarity Western ECL Substrate; Bio-Rad). Protein bands were visualized using the Molecular Imager ChemiDoc^TM^ XRS+ System (Bio-Rad). The integrated optical densities of the bands were quantified using the ImageJ software (NHI).

### 2.8. Spleen Cell Culture

Spleen fragments were collected and homogenized in ice-cold phosphate-buffered saline (PBS). Red blood cells were lysed with buffer containing NH_4_Cl, and the remaining cells were washed with RPMI. After washing, the suspension was centrifuged, and the cells were resuspended in 1 mL of complete medium. The concentration was adjusted to 5.0 × 10^6^ cells/mL, as determined by 0.1% trypan blue staining. Subsequently, the cells were challenged with *P. brasiliensis* exoantigen *Pb*Ag (20 mg/mL; Pb113 strain) or medium. The cultures were incubated at 37 °C and 5% CO_2_ in a humidified chamber for 24 h. Cell-free supernatants were harvested and stored at –80 °C for cytokine analysis.

### 2.9. Cytokine Measurements

The concentrations of TNF-α, MIP-1α, IL-10, IFN-γ, and TGF-β1 were measured via enzyme-linked immunosorbent assay using the DuoSet kit (R&D Systems, Minneapolis, MI, USA), according to the manufacturer’s instructions. The results were normalized to the protein concentrations of the lung extracts. For cell culture assays, the results were expressed as pg/mL.

### 2.10. Recovery of Viable Fungi

Lung samples were weighed and macerated in 1.0 mL of sterile PBS, and 100 μL was spread on BHI culture plates medium supplemented with 4% horse serum and 1% gentamicin using a Drigalski T loop. This protocol was performed in duplicate. The plates were sealed and incubated at 35 °C for 2 weeks. The number of colony-forming units was normalized per gram of tissue and log_10_ transformed.

### 2.11. Statistical Analysis

All statistical analyses were performed using the GraphPad Prism software (version 4.0; GraphPad Software, La Jolla, CA, USA). Analysis of variance (ANOVA) with Tukey’s post hoc test was performed to compare more than two independent variables. Statistical significance was set at *p* < 0.05.

## 3. Results

### 3.1. Silymarin in Association with CMX Ameliorates Pulmonary Fibrosis

We first investigated the development of PF after 12 weeks of treatment by analyzing the lung paraffin sections via Sirius red staining, followed by a quantitative assessment of the area with collagen fiber deposition. In the antifungal monotherapy, *Pb*-infected mice showed higher deposition of collagen fibers than untreated and infected mice (Figure 1A,D). In the combined silymarin/CMX treatment, *Pb*-infected mice displayed ameliorated PF with a significantly reduced area and deposition of collagen fibers (Figure 1A,E). The total lung collagen content was measured by quantifying the content of hydroxyproline, an amino acid abundantly found in collagen fibers. *Pb*-infected and silymarin/CMX-treated mice showed significantly lower hydroxyproline levels than *Pb*-infected and CMX-monotherapy-treated mice (Figure 1B). No changes in the fungal load were observed. In addition, sharp alterations with respect to the number and size of the granulomas were not observed among the groups.

### 3.2. Combined Silymarin/CMX Therapy Decreases Pulmonary Expression of Fibronectin and Increases MMP-2

Next, we quantified the pulmonary expression of proteins involved in wound-healing mechanisms. In *Pb*-infected mice, silymarin treatment and combined silymarin/CMX therapy induced lower fibronectin expression (Figure 2A) without interfering with α-SMA expression (Figure 2B). Furthermore, the combined silymarin/CMX treatment induced high expression of MMP-2 in *Pb*-infected mice (Figure 2C). No changes were observed in the expression of MMP-9 (Figure 2D) and TIMP-2 (Figure 2E).

### 3.3. Combined Silymarin/CMX Therapy Decreases the Pulmonary Levels of TGF-β1 and Increases IFN-y

The concentration of cytokines in the lung homogenates was also evaluated. We observed that *Pb*-infected cells showed high concentration of TGF-β1 regardless of treatment (Figure 3A). Particularly, CMX monotherapy induced high concentrations of TGF-β1 (Figure 3A). *Pb*-infected mice treated with combined silymarin/CMX showed a lesser amount of TGF-β1 than *Pb*-infected mice treated with CMX (Figure 3A). Furthermore, combined silymarin/CMX therapy resulted in a high concentration of IFN-γ-(Figure 3B). In general, *Pb*-infected mice exhibited high concentrations of TNF-α, MIP-1α, and IL-10; however, no significant differences were observed among treatments.

To evaluate whether the treatments affected leukocyte activity, we measured the production of TGF-β1 and IFN-γ in isolated spleen cells in response to *Pb* exposure. In cell culture studies, we observed that *Pb*-infected mice showed a high production of TGF-β1 except in mice treated with combined silymarin/CMX (Figure 3C). High levels of IFN-γ were detected in *Pb* antigen-stimulated spleen cells from *Pb*-infected mice treated with combined silymarin/CMX (Figure 3D).

## 4. Discussion

In general, the long-term antifungal treatment of PCM poses a complex challenge for clinicians and patients. Clinical evaluation of signs and symptoms must be performed periodically as well as the investigation of possible side effects, monitoring of antifungal medication compliance, evaluation of recovery of antigen-specific cellular immunity without an accurate laboratory test, assessment of possible relapses, and establishment of the best conduct for the symptomatic treatment of sequelae, such as PF and Addison’s syndrome, which must be permanent [4,5]. In the context of pulmonary sequelae, the present study shows that silymarin is a promising candidate in controlling PF. We observed that *Pb*-infected mice treated with combined silymarin and CMX therapy showed a reduction in the deposition of collagen fibers in the pulmonary area and a lower concentration of OH-proline compared to the untreated and infected mice. Our findings were associated with a decrease in TGF-β1 and fibronectin expression. In addition, we observed an increased expression of MMP-2 and IFN-γ.

The role of TGF-β in the development of lung fibrosis has been recognized for decades [22]. TGF-β is one of the most potent inducers of extracellular matrix (ECM) production, including collagen and other matrix proteins, and its expression is elevated in both animal models of lung fibrosis and fibrotic human lungs [23]. In the lungs, TGF-β is produced by a wide variety of cell types, including fibroblasts, myofibroblasts, alveolar macrophages, activated alveolar epithelial cells, endothelial cells, and neutrophils [24]. In patients with PCM, enhanced expression of TGF-β1 has been detected in fibrotic tissues [25] and is highly secreted by monocytes ex vivo [6]. The involvement of TGF-β1 in PF has also been demonstrated in several murine models of PCM, and TGF-β1 has been used as an important parameter in determining the antifibrotic effect of new therapeutic approaches for *Paracoccidioides*-induced PF [12]. The protective effect of silymarin on TGF-β1 production has also been observed in a CCl_4_-induced liver fibrosis model [26]. Silymarin also inhibited the progression of fibrosis in the early stages of liver injury in CCl_4_-treated rats and was associated with reduced hepatic expression of TGF-β1 [27]. In an HCl-induced acute lung injury model, silymarin treatment induced a detectable decrease in the number of TGF-β1-positive cells in the lungs [28]. Lipopolysaccharide and yeasts of *Pb*-stimulated human monocytes treated with silibinin, the most prevalent component of the silymarin complex, released lower levels of TGF-β1 than untreated stimulated cells [18]. In the present study, we also observed a significant reduction in TGF-β1 secretion by *Pb* antigen-stimulated spleen cells treated with silymarin.

Fibronectin is another important mediator involved in the process of PF. Fibronectin is known to act as a chemoattractant for fibroblasts [29] or monocytes [30]. Furthermore, fibronectin or its fragments may act as mitogens for fibroblasts [31,32]. Gray et al. [33] demonstrated that the breakdown of fibrinogen products or elements of the pathway by which fibrinogen forms fibrin may also be mitogenic in fibroblasts. Fibronectin and other ECM proteins act as substrates for the attachment of *P. brasiliensis* during host–pathogen interactions (adaptation, adhesion, and invasion), with a direct impact on fungal virulence and host immune response [34,35]. However, the role of fibronectin has not been evaluated in the context of PCM-PF. In our experimental model, we observed an elevated expression of this ECM in the lungs of *Pb*-infected mice. In addition, combined silymarin/CMX therapy induced lower expression of fibronectin in the pulmonary tissue. In a recent report, Ali et al. [36] showed that silibinin alleviated silica-induced PF, harnessed the tissue architecture, and modulated epithelial-mesenchymal phenotypic alterations. Silymarin treatment in mice with CCl_4_-induced liver injury downregulated the expression of fibrosis-related genes α-SMA and fibronectin [37]. In the present study, immunoblotting was used to quantify α-SMA expression; however, no changes were observed. An experimental PF model induced by bleomycin also demonstrated no alterations in α-SMA expression in fibrotic lungs cells [38]. Nevertheless, further studies are necessary to investigate whether this well-known fibrotic marker is altered in the context of PCM-PF.

Collagen degradation pathways have been addressed in some studies on PCM [39,40,41] since they exert a potentially remarkable effect given the rapid renewal in the normal lung during the infectious process and in PF [40,42]. ECM degradation is primarily controlled by MMPs, a family of secreted zinc-dependent enzymes that are collectively capable of degrading the main components of the matrix [43]. MMPs are regulated at the gene transcription level through activation of a latent proenzyme and are inhibited by a family of secreted proteins known as tissue matrix metalloproteinase inhibitors (TIMPs), which bind to the active site and specific latent forms of MMP [44]. In an experimental PF model induced by bleomycin, the expression of MMP enzymes and their TIMP inhibitors were observed to be selective and were temporally different during the development of fibrosis [45]. In general, induction of PF with bleomycin as well as in *P. brasiliensis* infection [41] results in upregulation of MMP and TIMP gene expression in lung tissue. The strong correlation between increased expression of ECM genes, differential expression between MMPs and TIMPs genes, and histopathological evidence of fibrosis suggests that dysregulation in matrix remodeling may contribute to the pathology of PF. In this context, the enhanced expression of MMP-2 in the lungs of *Pb*-infected mice treated with combined silymarin/CMX may indicate other possible mechanisms underlying the effect of silymarin on PF. In general, decreased activity of gelatinases, especially MMP-2, is related to the development of fibrosis, probably due to a decrease in the capability of ECM remodeling [46,47,48]. However, different models of fibrosis treated with silymarin have shown both upregulation and downregulation of MMP-2 expression [49,50,51]. Thus, the expression of different molecules involved in collagen degradation pathways may vary over time, and further temporal analyses may be useful in understand this process.

Fibrotic diseases and the Th1/Th2 paradigm have been extensively addressed in the literature. In the conceptual model proposed by Wynn [52], the Th1-cell cytokine IFN-γ directly controls collagen synthesis by fibroblasts, which in turn regulates the balance of MMP and TIMP expression, thereby controlling the rates of collagen degradation and synthesis, respectively. In addition, IFN-γ and IL-12 may inhibit fibrosis by reducing pro-fibrotic cytokine expression by Th2 cells, which are responsible for the release of IL-4, IL-5, and IL-13, which enhance collagen deposition through various mechanisms. In the present study, we observed that *Pb*-infected mice treated with combined silymarin and CMX had increased concentrations of IFN-γ in the lungs, which was also observed in the spleen cell culture. This finding reinforces the antifibrotic role of the combined therapy.

IFN-γ is also directly involved in the killing of *P. brasiliensis* by activate phagocytes [53]. However, we did not observe a decreased recovery of viable fungi, as expected. CMX is a fungistatic drug widely used in Brazil to treat PCM and is the first choice to treat neuroparacoccidiodomycosis [5]. In our previous study, we also did not observe changes in the fungal burden of *Pb*-infected mice treated with combined antifibrotic and antifungal therapy after eight weeks of treatment [11]. In the present study, we extended the treatment time; however, no changes were observed. We observed a decrease in *P. brasiliensis* load after 20 weeks of CMX treatment (unpublished data). Silibinin-encapsulated human monocytes infected with *P. brasiliensis* resulted in the suppression of pro-inflammatory cytokines and nitric oxide production but did not affect fungicidal activity [18]. Further studies using itraconazole as an antifungal treatment are necessary to validate our findings [12,54].

This study had several limitations. The antifibrotic effect of combined silymarin/CMX therapy was based only on the concentration of silymarin as well as at one time point. Other methodologies, such as immunohistochemistry or real-time PCR, could help us confirm some data that were not clearly observed using Western blotting.

## 5. Conclusions

In conclusion, the present study showed that oral administration of silymarin may improve pulmonary fibrosis induced by *Pb*-infected mice via a decrease in fibrotic parameters, including OH-proline, deposition of collagen fibers, and modulation of components of the wound-healing process. The mechanism underlying the activity of silymarin in the treatment of PF in PCM could be explored in the future.

## Figures and Tables

**Figure 1 jof-08-01010-f001:**
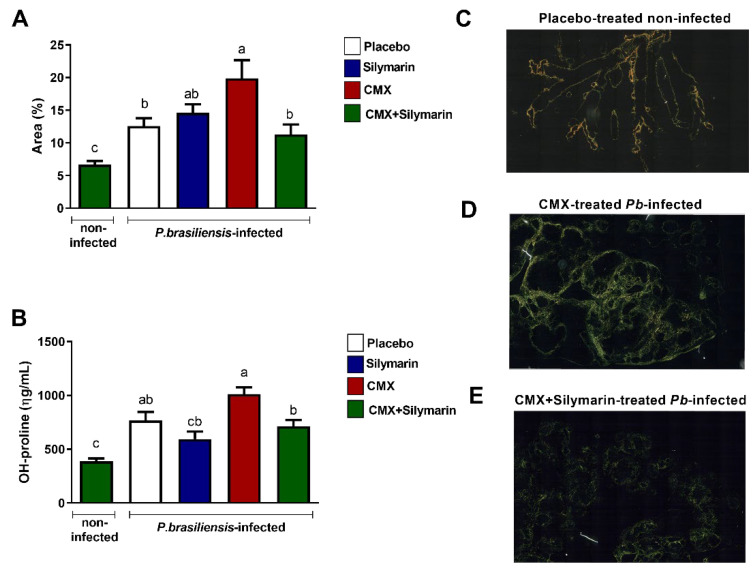
Evaluation of the effects of silymarin in combination with the antifungal cotrimoxazole (CMX) on pulmonary fibrosis parameters. BALB/c mice were infected with *P. brasiliensis*, and at the 8th week of infection, silymarin/CMX schedule and placebo were started. The mice were evaluated after 12 weeks of treatment. (**A**) Determination of the percentage of collagen fibers in the lungs. (**B**) Quantification of hydroxyproline in the lungs. Values are expressed as ηg/mL. (**C**) Section of a normal lung, stained with Sirius red, submitted to polarized light microscopy that was used to visualize birefringent collagen fibers. (**D**) Section of a CMX-treated *P. brasiliensis*-infected mice, (**E**) Section of silymarin/CMX-treated *P. brasiliensis*-infected mice. ANOVA with Tukey post hoc test; the values with different superscript letters in a column are significantly different (*p* < 0.05).

**Figure 2 jof-08-01010-f002:**
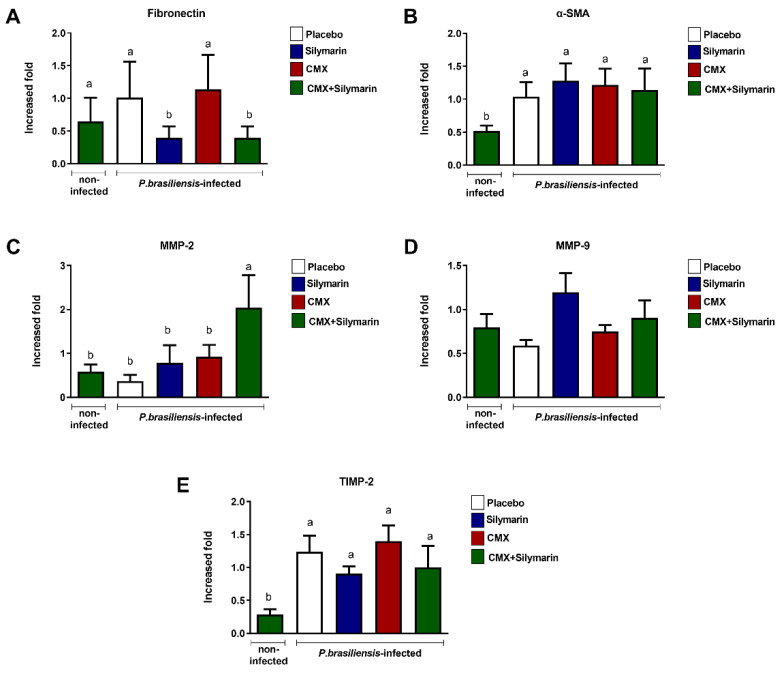
Influence of silymarin in combination with the antifungals cotrimoxazole (CMX) on the expression of components of extracellular matrix BALB/c mice were infected with *P. brasiliensis*, and at the 8th week of infection, silymarin/CMX schedule and placebo were started. The mice were evaluated after 12 weeks of treatment for: (**A**) fibronectin, (**B**) α-SMA, (**C**) MMP-2, (**D**) MMP-9, (**E**) and TIMP-2. ANOVA with Tukey post hoc test; the values with different superscript letters in a column are significantly different (*p* < 0.05); *n* = 5–7.

**Figure 3 jof-08-01010-f003:**
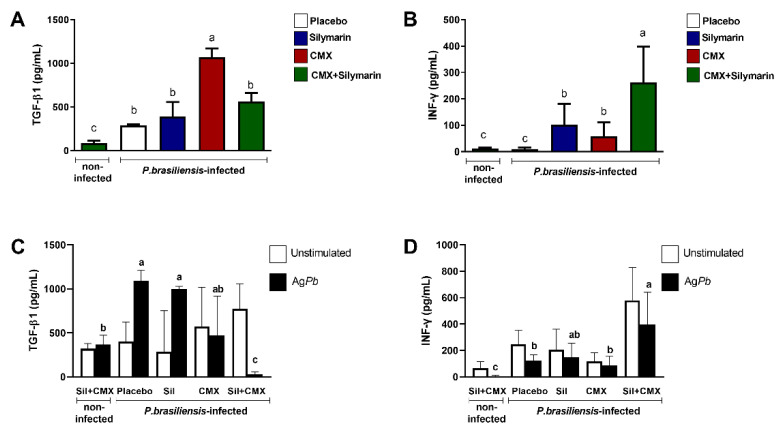
Influence of silymarin in combination with the antifungals cotrimoxazole (CMX) on the quantification of TGF-β1 (**A**) and IFN-y (**B**) and release of TGF-β1 (**C**) and IFN-y (**D**) by spleen cells stimulated or not with *P. brasiliensis* exoantigen. BALB/c mice were infected with *P. brasiliensis*, and at the 8th week of infection, silymarin/CMX schedule and placebo were started. The mice were evaluated after 12 weeks of treatment. ANOVA with Tukey post hoc test; the values with different superscript letters in a column are significantly different (*p* < 0.05); *n* = 5–7.

## Data Availability

All relevant data are within the paper.

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
