# Peer review of "Combined Silymarin and Cotrimoxazole Therapy Attenuates Pulmonary Fibrosis in Experimental Paracoccidioidomycosis"

_jof, 2022, doi:10.3390/jof8101010_

Round 1

Reviewer 1 Report

Comments to the authors:

The paper is very interesting in the field. Authors describes that the use of silymarin and cotrimoxazole combined therapy attenuates pulmonary fibrosis in experimental paracoccidioidomycosis. The paper is well written and the report is sound, but I do think that minor issues need to be dealt with prior to publication:

Page 6, Line 183, the authors state “We first investigated the development of PF after 12 weeks of treatment, analyzing liver paraffin sections by Sirius red staining followed by a quantitative assessment of the area with collagen fiber deposition”, it is confusing, the analyzes were they done in the lung or in the liver?

In the legend of figure 3, it is important to place the value of P.

Also important is the number of animals used in each experiment, how many replicates, and how many times the experiments were repeated.

Is there any difference in the type of inflammatory response, if the treatment affected the number and size of granulomas?

Author Response

REVIEWER #1. Page 6, Line 183, the authors state “We first investigated the development of PF after 12 weeks of treatment, analyzing liver paraffin sections by Sirius red staining followed by a quantitative assessment of the area with collagen fiber deposition”, it is confusing, the analyzes were they done in the lung or in the liver?.

Authors’ response: Thank you for the comment and sorry about that. The analyzes were performed in lungs. The phrase was re-written and the word correctly changed (line 185).

REVIEWER #1. In the legend of figure 3, it is important to place the value of P.

Authors’ response: Thank you for the comment. We have added the p value, as requested (lines 242-243).

REVIEWER #1. Also important is the number of animals used in each experiment, how many replicates, and how many times the experiments were repeated.

Authors’ response: Thank you for the comment. Each group of mice was composed of seven animals, that were inoculated with P. brasiliensis, by i.t. route. In each set of experiments, successful inoculation was confirmed via histopathological evaluation. Failed inoculation was considered if no infectious injury was identified in the lung tissue; thus, the mice were not included in the analyses. All experiments were per-formed in duplicate, as recommended by the ethical committee. We have added this information in the manuscript. (Lines 103-115)

REVIEWER #1. Is there any difference in the type of inflammatory response, if the treatment affected the number and size of granulomas?

Authors’ response: Thank you for the comment. In the present study we did not perform quantitative analyses of number and size of granulomas. In general, In addition, sharp alterations with respect to the number and size of the granulomas were not observed among the groups. We have added this information in the manuscript. (Lines 194-195)

Reviewer 2 Report

  • A brief summary In the present study, authors propose a new therapeutic approach for pulmonary fibrosis in paracoccidioidomycosis (PCM) including herbal formulations and plant bioactive molecules as potential candidates. It is important issue as there is no available pharmacological treatment of pulmonary sequels for PCM patients, althought experimental studies have identified promising therapeutic strategies including drugs used for idiopathic pulmonary fibrosis, immunotherapy, antibody-based therapy, cellular therapy, and vaccination.
  • General concept comments
    Methodology is well described, results clearly presented and discussion comprehensive. The findings of the study are important and relevant because this area of PCM treatment is still not solved and the results of the study opens the area for other investigators' groups. 
  • Specific comments Very well written

Author Response

Thank you for the kind comment
